

# Gua Sha, a press-stroke treatment of the skin, boosts the immune response to intradermal vaccination

Tingting Chen[1,*], Ninghua Liu[1,*], Jinxuan Liu[1], Xiaoying Zhang[1], Zhen Huang[1], Yuhui Zang[1], Jiangning Chen[1], Lei Dong[1], Junfeng Zhang[1,2] and Zhi Ding[1,2]

[1] State Key Laboratory of Pharmaceutical Biotechnology, School of Life Sciences, Nanjing University, Nanjing, Jiangsu, China
[2] Collaborative Innovation Center of Chemistry for Life Sciences, Nanjing University, Nanjing, Jiangsu, China
[*] These authors contributed equally to this work.

## ABSTRACT

**Objective**. The skin is an important immunological barrier of the body as well as an optimal route for vaccine administration. Gua Sha, which involves press-stroke treatment of the skin, is an effective folk therapy, widely accepted in East Asia, for various symptoms; however, the mechanisms underlying its therapeutic effects have not been clarified. We investigated the influence of Gua Sha on the immunological features of the skin.

**Methods**. Gua Sha was performed on BALB/c mice and the effects were evaluated using anatomical, histological, and cytometric methods as well as cytokine determination locally and systemically. The effect on intradermal vaccination was assessed with antigen-specific subtype antibody responses.

**Results**. Blood vessel expansion, erythrocyte extravasation, and increased ratios of immune active cells were observed in the skin tissue following the treatment. Pro-inflammatory cytokines were up-regulated, and immunosuppressive cytokines, down-regulated, in the treated and untreated skin and systemic circulation; no obvious variations were detected in case of anti-inflammatory cytokines. Interestingly, intradermal delivery of a model vaccine following Gua Sha induced about three-fold higher IgG titers with a more Th1-biased antibody subtype profile.

**Conclusion**. Gua Sha treatment can up-regulate the innate and adaptive immune functions of the skin and boost the response against intradermal antigens. Thus, Gua Sha may serve as a safe, inexpensive, and independent physical adjuvant for intradermal vaccination.

Corresponding authors
Junfeng Zhang, jfzhang@nju.edu.cn
Zhi Ding, dingzhi@nju.edu.cn

## INTRODUCTION

People scratch their skin unconsciously in response to an itching sensation despite the fact that it may open and damage wounds at the site and result in infection and even inflammation. Scratching can accelerate the expansion of erythema and exacerbate symptoms in patients with contact urticaria or atopic dermatitis (*Wuthrich, 1998*). With these side effects as tradeoffs, scratching must have some value that allows its existence

over time, besides offering relief from itching and the dissipation of invading insects and parasites. Scratching might be able to extend signals locally to the skin and modulate defensive functions. The Chinese invented therapies based on mechanical manipulation of the skin, e.g., Gua Sha and Baguan (Cupping), around 2,000 years ago, which are empirically effective towards multiple conditions such as chronic pain, common cold, heatstroke, and respiratory problems. The functions of scratching and the mechanisms underlying the therapeutic effects of Gua Sha encouraged us to undertake this study.

Literally, Gua refers to the scratching of the skin, while Sha refers to the petechiae and texture appearing after scratching (*Odhav et al., 2013*). Gua Sha is defined as repeated, unidirectional, press-stroke of the lubricated skin area with a smooth-edged instrument (Figs. 1A and 1B) until Sha, i.e., blemishes, appear due to blood congestion (Fig. 1C). The blemishes fade and completely resolve within 2–5 days in humans with the symptoms getting alleviated immediately or few hours later (*Braun et al., 2011*). The skin is the largest organ of the body, the interface most exposed to the external environment, and the first-line defense against a broad range of microorganisms. At present, it is clear that the skin serves as a highly sophisticated, potent immune surveillance system related to both innate and adaptive immunity (*Weniger & Glenn, 2013*). Its immunological functions primarily rely on the Langerhans' cells (LCs) in the epidermis and the dermal dendritic cells (DCs) in the dermis (*Merad, Ginhoux & Collin, 2008*), which are involved in antigen capturing, processing, and presentation, during the skin barrier disruption, pathogen invasion, or vaccination; and can cause inflammation, immune activation, or tolerance, depending on various conditions (*Matzinger, 2002*; *Ding et al., 2010*; *Engelke et al., 2015*). Therefore, the therapeutic mechanism of Gua Sha is believed to be highly relevant for the immunological functions of the skin.

To the best of our knowledge, the effects of Gua Sha treatment on the immunological features of the skin have not been clarified. In the current study, it is hypothesized that Gua Sha-induced extravasation of blood and controllable skin tissue damage leads to the wound-healing process, including the increase in the level of pro-inflammatory cytokines, and decrease in the level of immunosuppressive cytokines. This results in sensitized innate and adaptive immunity, both locally and systemically. Our studies helped to establish a connection between Gua Sha and the immunological features of the skin. The effect of this treatment on the surface microcirculation in the skin tissue was also confirmed. The skin cytokine levels post-Gua Sha as well as the antibody titers after vaccine administration at the treatment site were determined in preclinical trials. Thus, the effects of Gua Sha on the skin immune system as well as the intradermal vaccination are being studied.

## MATERIALS AND METHODS

### Materials

Ovalbumin (OVA) and Freund's incomplete adjuvant (FIA) were purchased from Sigma-Aldrich (Shanghai, China). Pentobarbital sodium was obtained from Merck, and Tween 20 from Sangon Biotech Co., Ltd (Shanghai, China). Horseradish peroxidase

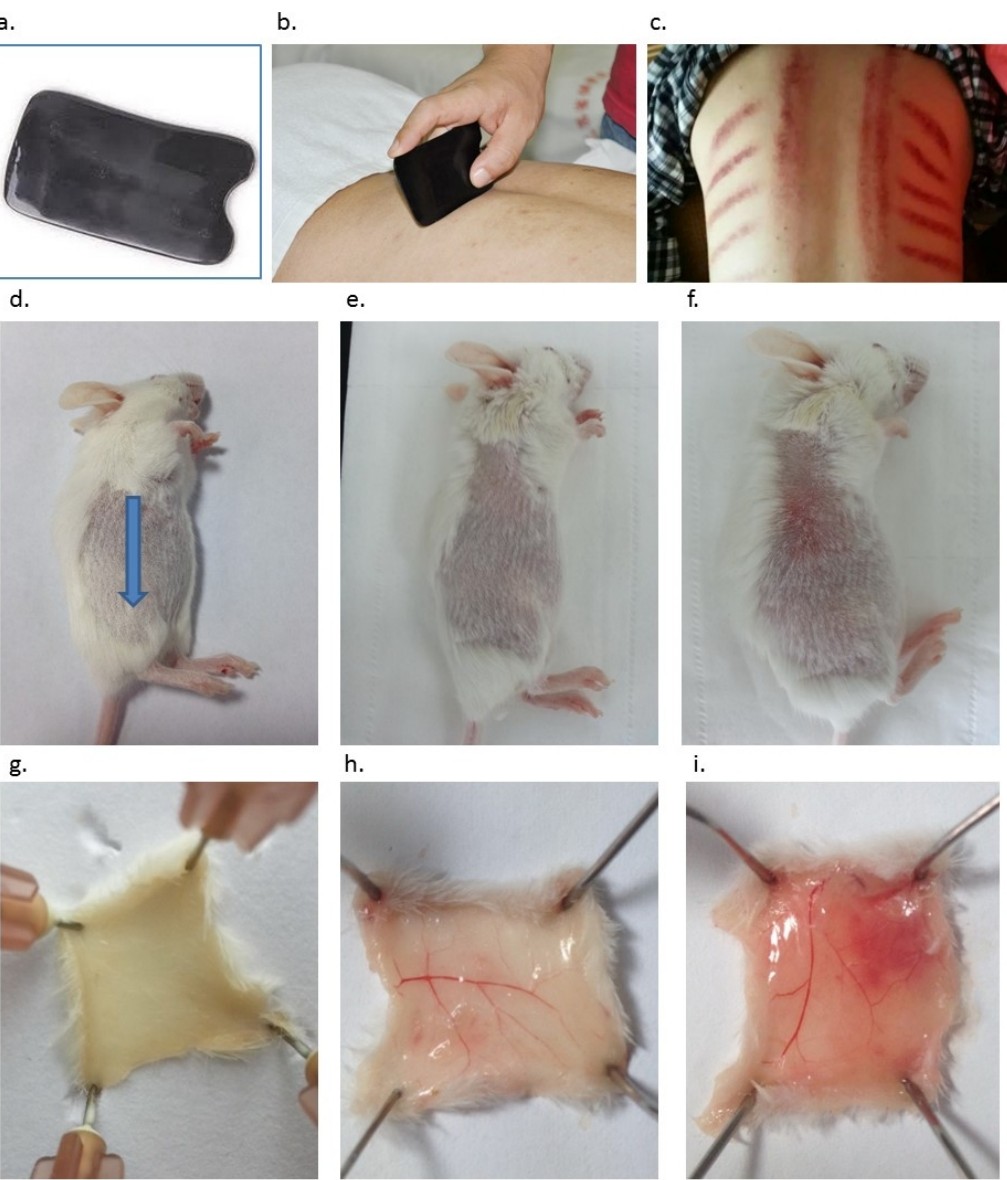

**Figure 1** **Introduction of Gua Sha and representative anatomic images of Gua Sha treatment on mouse skin.** (A) The smooth-edged instrument made of bull-horn for Gua Sha treatment, with the size and shape similar as a credit card. (B) Gua Sha treatment on a person's back. (C) The pattern of blemishes resulted from Gua Sha treatment in human. Mouse skin prior to Gua Sha treatment observed from the *stratum corneum* side (D), (blue arrow indicates the direction of Gua Sha operation) and the dermal side (G); mouse skin after 20 or 40 scrapes observed from the *stratum corneum* side (E & F) and the dermal side (H & I) . Photos were taken 30 min after Gua Sha treatment from the *stratum corneum* side, then the mice were euthanized for observation from the dermal side. Images are representative ones from three mice per group.

(HRP)-conjugated goat anti-mouse IgG ($\gamma$-chain specific), IgG1 ($\gamma$1-chain specific), and IgG2a ($\gamma$2a-chain specific) were purchased from Southern Biotech (Birmingham, USA). A chromogen, 3,3′,5,5′-tetramethylbenzidine (TMB), and the substrate buffer were purchased from Beyotime B.V. (Shanghai, China). Skim milk powder (Yili) was obtained

from the local supermarket. All other chemicals were of analytical grade, and all solutions were prepared with Milli-Q water.

## Animals

Female BALB/c mice (H-$2^d$), 6–8 weeks old at the start of the experiments, were purchased from the Experimental Animal Center of Nanjing Medical School (Nanjing, China) and maintained under standardized pathogen-free conditions in the animal facility of the State Key Laboratory of Pharmaceutical Biotechnology, Nanjing University. All mice were housed in standard cages, with free access to food and water. The animals were maintained at a constant temperature (20–21 °C) and humidity (55% ± 5%), with a 12-h light/dark cycle. All animals were handled in accordance with the Declaration of Helsinki, 1975 (as revised in 2008) concerning animal rights, and the protocols were approved by the Institutional Animal Care and Use Committee of Model Animal Research Center of Nanjing University (Ref No. 20130015). Efforts were made to minimize the amount of animals used and ameliorate animal suffering.

## Gua Sha treatment on experimental mice

Mice were anesthetized by intraperitoneal injection of 100 mg/kg pentobarbital. Gua Sha was performed on the side of the mouse's back by using a smooth-edged instrument made of bull-horn (Figs. 1A and 1D), after the hair was shaved with a clipper a day prior to the experiment. This material was chosen instead of plastic to avoid accumulation of static electricity during operation. The shaved skin area was wiped with 70% ethanol and left to dry. Then, press-stroke treatment (called "scrape" henceforth) was applied 20 or 40 times in a unidirectional manner, with an angle of ∼90° between the instrument and the mouse's back. The force of scrape was optimized and standardized at ∼3 N, so that blemishes would appear between 20 and 40 scrapes, while the inner organs would not be damaged. Twenty scrapes took ∼30 s, and a skin area of ∼3 cm$^2$ was treated. Lubricant oil was not included in the experiments, as the friction was not harmful and the effect of lubricant oil on the skin tissue and vaccine delivery would need to be excluded. No abrasion-induced open wound was visible during and after the treatment, and the skin remained intact.

## Histological imaging of the treated skin

Mice were euthanized by cervical dislocation at the indicated time points after Gua Sha treatment. For histopathological examination, samples of treated/untreated skin tissue were obtained, fixed in Bouin's buffer, embedded in paraffin, and sectioned. The sections stained with Masson's trichrome method were analyzed using optical microscopy (Nikon TE2000-U; Nikon, Tokyo, Japan) (*Goldner, 1938*).

## Flow cytometry analysis of the treated skin

At indicated time points after Gua Sha treatment, the mice were euthanized by cervical dislocation. Approximately 3-cm$^2$ skin tissue from the treatment site was excised, chopped to small pieces, and digested in 4 mg/ml type I collagenase in Hank's balanced salt solution (HBSS; Sigma-Aldrich) at 37 °C for 2 h. Single cell suspensions were

prepared using a 70-µm cell strainer (BD Bioscience, Shanghai) at the concentration of $1 \times 10^7$/ml in phosphate-buffered saline (PBS) containing 1% BSA. Cells were stained using APC-conjugated hamster anti-mouse CD11c, PE-conjugated rat anti-mouse F4/80 (BioLegend), or their corresponding isotype controls. After incubation on ice for 30 min, the cells were washed twice with PBS containing 1% BSA and re-suspended for flow cytometry analysis using BD FACS Calibur. Data were analyzed using FlowJo software (Tree Star).

## Cytokine profile analysis

Mice were euthanized by cervical dislocation at indicated time points after Gua Sha treatment. Approximately 100 mg of skin tissue from the treatment site was excised and rinsed in cold PBS (at 1 mg/µl), and then homogenized on ice using Lysing Matrix D tubes and Fast Prep-24 homogenizer (MP Biomedicals, Santa Ana, CA). Five groups of skin samples were obtained as follows: (1) untreated skin from naïve mice; (2) treated skin samples taken 1 h after 40 scrapes; (3) untreated skin samples from treated mice 1 h after 40 scrapes; (4) treated skin samples taken 2 h after 40 scrapes; (5) untreated skin samples from treated mice 2 h after 40 scrapes. Supernatants were collected after centrifugation at 14,000 rpm for 20 min at 4 °C. Cytokine levels, including tumor necrosis factor (TNF)-$\alpha$, interleukin (IL)-1$\beta$, IL-4, IL-5, IL-6, IL-10, IL-12p70, IL-13, and IL-23 of the lysates were quantified using enzyme-linked immunosorbent assay (ELISA) kits, following the manufacturer's instructions (4A Biotech Co. Ltd., Beijing, China). Blood samples, 4–5 drops each time per mouse, were collected from the retro-orbital venous sinus of the mice 0.5 and 1 h after 40 scrapes. Blood samples of the untreated mice were collected with the same procedure. Cell-free serum was obtained in collecting tubes with coagulation agent and separation gel (Gongdong, Taizhou, China) by centrifugation after clot formation. TNF-$\alpha$, IL-1$\beta$, and IL-6 levels in the sera were also determined and compared to those of the untreated mice. Serum nitric oxide level was determined using an assay kit (Jiancheng Bioengineering Institute, Nanjing, China) based on the Griess reaction method, following the manufacturer's instructions (*Guevara et al., 1998*).

## Immunization and serum antibody assays

Mice were intradermally (*i.d.*) immunized with 5 µg of OVA under anesthesia three times on day 1, 21, and 42 at a site between the back and thigh 10 min after 20 or 40 scrapes, and euthanized on day 56. Groups of OVA alone and OVA with FIA delivered *i.d.* to naïve mice were included as the negative and positive controls, respectively. Blood samples were drawn from the tail vein one day before each immunization or from the retro-orbital venous sinus during euthanasia under systemic anesthesia. Cell-free sera were obtained as mentioned above and stored at −80 °C.

OVA-specific serum IgG, IgG1, and IgG2a titers were determined by ELISA as described previously (*Ding et al., 2009*). Briefly, ELISA plates (Costar 9018; Elisa, Shanghai, China) were coated with OVA at 4 °C overnight. Two-fold serial dilutions of the serum samples were applied, and the antibodies were detected by HRP-conjugated goat anti-mouse IgG, IgG1, or IgG2a by using TMB as the substrate. Antibody titers are expressed

as the calculated sample dilution times corresponding to the half of the maximum absorbance at 450 nm of a complete sigmoid absorbance-log dilution curve. If the samples were not diluted in the optimal range, additional measurements with more diluted or concentrated samples were performed to complete the $S$-shaped curve. Mice whose serum samples did not reach the half-saturated absorbance value at the lowest (ten-fold) dilution were considered non-responders at that time point and the titers were arbitrarily considered 10.

### Statistical analysis

Antibody titers were logarithmically transformed for better normality. Analysis was performed as indicated. Statistical analysis was carried out using Graphpad (Prism, San Diego, CA, USA) and a $p$ value of $< 0.05$ was considered significant.

## RESULTS

### Skin scrapes lead to blood congestion, blood vessel expansion, and infiltration of immune active cells locally

Treated skin samples were observed with the naked eye as well as with Masson's staining in order to study the effect of scrapes on the skin. The skin of the naïve mouse after hair removal looked white with a pinkish background. From the dermal side, it was milky white with hardly any capillaries visible (Figs. 1D and 1G). Scrapes were applied 20 or 40 times in a unidirectional manner on the mouse's back. When observed 30 min after treatment, the skin became darker from the *stratum corneum* side, with a few blood vessels distinguishable from the dermal side after 20 scrapes (Figs. 1E and 1H). After 40 scrapes, petechiae appeared on the *stratum corneum* side, and subcutaneous microvascular blood extravasation and bruises could be observed from the dermal side (Figs. 1F and 1I). The vessels in the subcutaneous tissue expanded considerably, with some of them being located closer to the *stratum corneum,* as shown in the images of Masson's staining of the Gua Sha-treated skin sections (Fig. 2). The increased diameter of the vessels indicated an enhanced blood and lymphatic flow, allowing more rapid substance exchange with the interstitial fluid. Red blood cells, probably together with other cell types and contents, dispersed through the ruptured peripheral blood vessels into the dermis and subcutaneous fat tissues, followed by accumulation for hours.

In the Gua Sha-treated skin tissue, the ratios of CD11c$^+$ cells, including DCs and activated T lymphocytes, and F4/80 macrophages were found to be increased, as observed by flow cytometry analysis (Fig. 3). In the untreated mouse skin, CD11c$^+$ cells accounted for ∼4% of the total cell population. This proportion increased to ∼9%, 12%, and 14% at 15, 30, and 60 min after treatment, respectively (Figs. 3A and 3B). A similar trend was observed in case of macrophages present in the treated skin tissue; the proportion of macrophages increased with time after Gua Sha treatment, from 16.5% to ∼20% (Figs. 3C and 3D), probably because of the infiltration from the expanded or ruptured blood vessels.

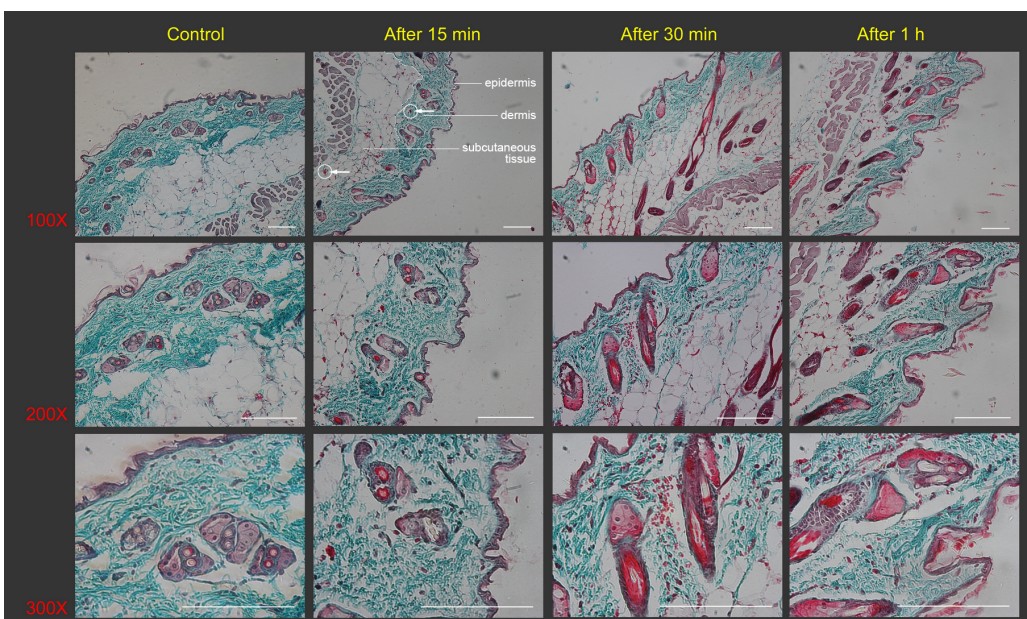

**Figure 2  Histological images of mouse skin sections prior to and after Gua Sha treatment.** Representative images of Masson's trichrome staining of mouse skin sections prior to and after 40 scrapes at different time points and magnifications were shown from three mice per group, scale bar = 100 µm. As an example, in the image (100×) taken 15 min after Gua Sha treatment, the boundaries between epidermis, dermis and subcutaneous tissue were depicted with white dots, and two groups of extravasated red blood cells were pointed out with white circles and arrowheads in dermis and subcutaneous tissue.

## Skin scrapes cause variations in the local and systemic levels of cytokines and nitric oxide

As Gua Sha treatment enhances microcirculation, more rapid substance exchange occurs among the blood, lymphatic fluid, interstitial fluid, and the infiltrated immune active cells. This changes the microenvironment of the treated skin tissue. The levels of the representative cytokine groups present locally in the treated skin tissue and systemically in the untreated skin area were determined and compared with those of naïve mice (Fig. 4). Local concentrations of most pro-inflammatory cytokines examined, including TNF-$\alpha$, IL-6, IL-12p70, and IL-23, increased moderately; but a significant increase was observed after the treatment, except IL-1$\beta$. Among them, TNF-$\alpha$ level increased in both treated and untreated skin tissues, while IL-6, IL-12p70, and IL-23 levels in the untreated skin area of the treated mice remained constant. The immunosuppressive cytokine IL-10 was present at lower levels in the treated skin tissue 1 h and 2 h after treatment, and in the untreated skin area 2 h after treatment, compared to that in the untreated mice, indicating an overall up-regulation of immune reactivity (Fig. 4D). The levels of anti-inflammatory cytokines IL-4, IL-5, and IL-13 in the skin tissues of treated and untreated mice were also examined, but no remarkable differences were detected (Fig. S1). In the serum samples of the treated mice, the levels of TNF-$\alpha$, IL-1$\beta$, and IL-6 increased significantly, compared to those from untreated mice (Figs. 5A–5C). Consistent with the anatomical and histological observations, nitric oxide content, which leads to vasodilation and increased blood flow,

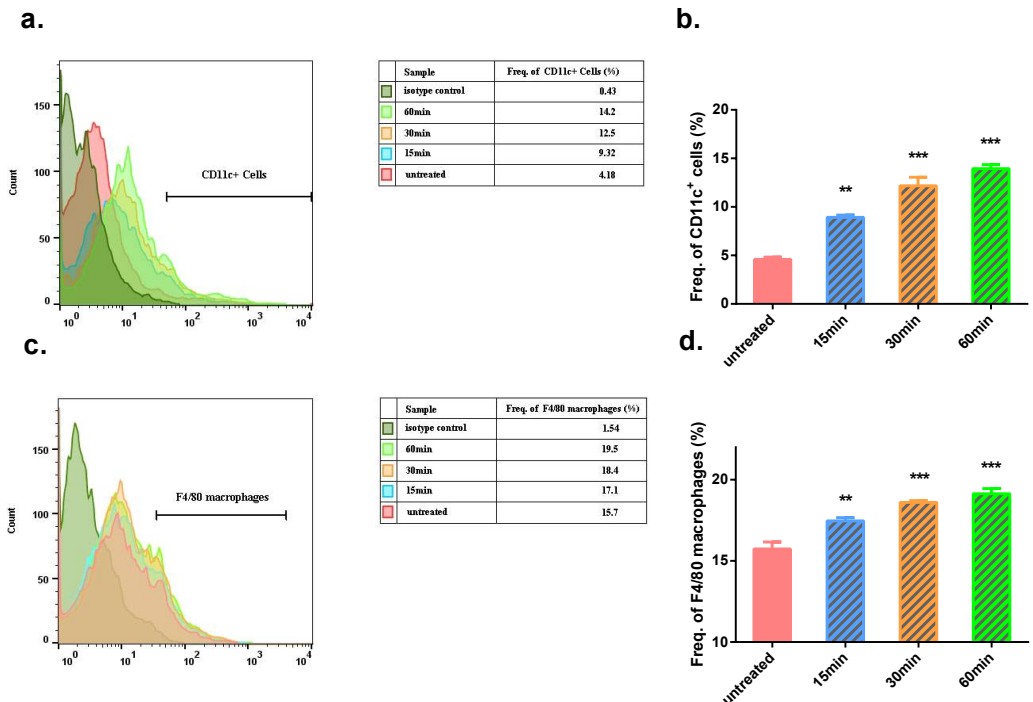

**Figure 3  Flow cytometry analysis of Gua Sha-treated skin tissue.** Single cell suspensions from Gua Sha-treated mouse skin or untreated control were prepared and stained with APC-conjugated anti-mouse CD11c and PE-conjugated anti-mouse F4/80. Representative histograms showing the counts of CD11c$^+$ cells and F4/80 macrophages from three mice per group were shown in (A) & (C), and the percentages of these cell populations were plotted in (B) & (D), respectively. Data were shown as Mean + SEM. Statistical comparisons were made between each treated group and the untreated group. ($n = 3$; *, $p < 0.05$; **, $p < 0.01$; ***, $p < 0.001$; One-way ANOVA with Dunnett's posttest.)

was significantly up-regulated (Fig. 5D). In addition to increasing the infiltration of immune active cells such as neutrophils, monocytes, and macrophages from the blood, nitric oxide enhances immune defense by acting as a free radical with oxidative pressure and is toxic to bacteria and intracellular parasites.

## Skin scrapes boost the immune response to intradermal OVA vaccination

Because the levels of some pro-inflammatory cytokines were up-regulated locally, it was hypothesized that the Gua Sha treatment would enhance adaptive immunity against intradermal pathogens. An *in vivo* vaccination study was performed to examine its effect on the immune defense of the skin. Intradermal injection, instead of dermal application, was chosen for vaccine administration to ensure exact vaccine dosage. The OVA-specific isotype antibody titers of the serum samples from three different time points were determined (Fig. 6). The OVA-specific IgG titers of the Gua Sha-treated groups did not differ significantly from those of the untreated group after prime (Fig. 6A). Interestingly while as expected, the systemic humoral response to OVA was augmented, as evidenced by the IgG titers after the first and second booster doses (Figs. 6B and 6C) and IgG1 titer after second booster dose (Fig. 6D). 20 and 40 scrapes of the skin induced about two and

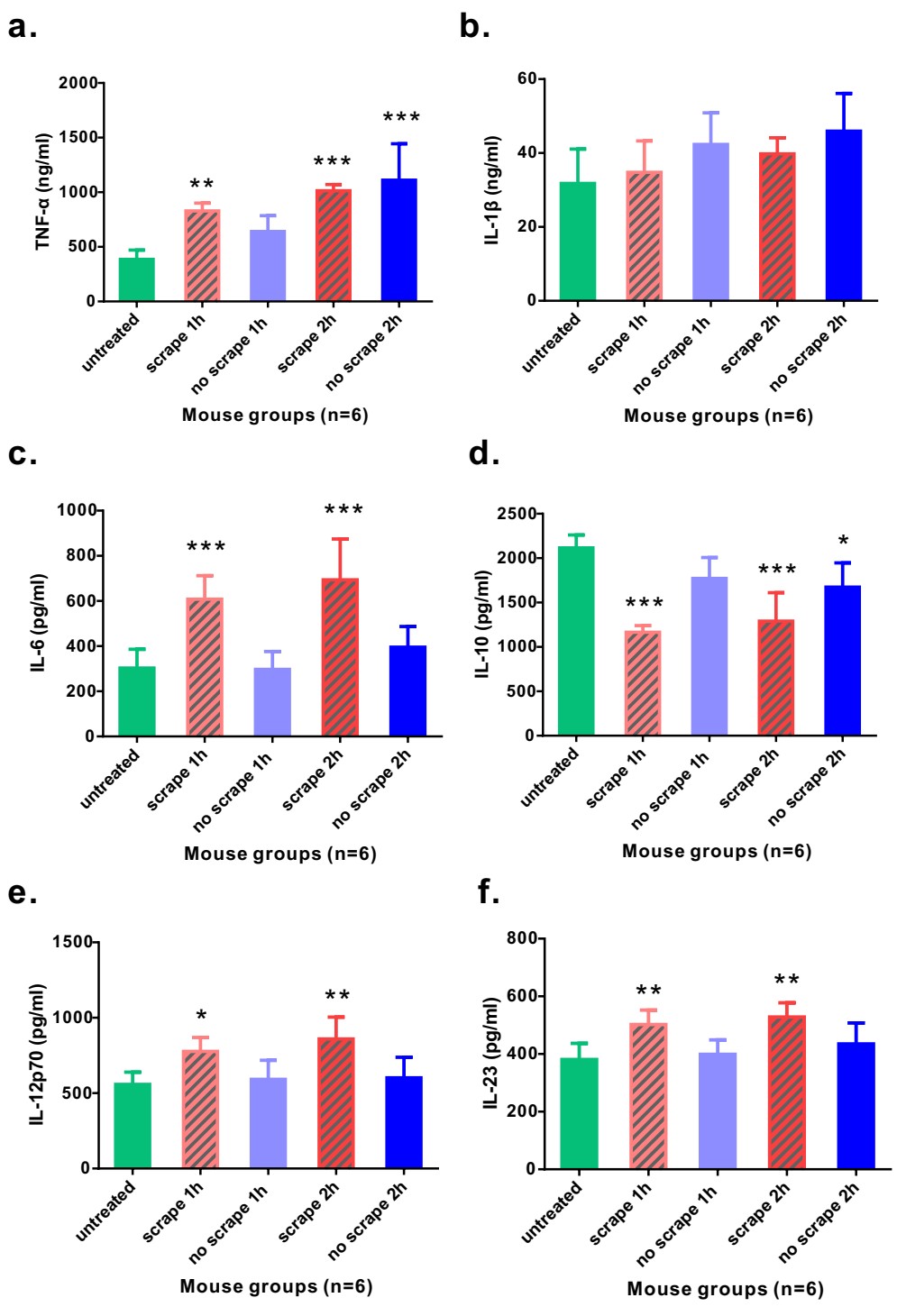

**Figure 4** **Levels of cytokines from skin tissues with or without 40 scrapes Gua Sha treatment.** Groups were set as following: (1) untreated skin from naïve mice; (2) treated skin samples taken 1 h after 40 scrapes; (3) untreated skin samples from treated mice 1 h after 40 scrapes; (4) treated skin samples taken 2 h after 40 scrapes; (5) untreated skin samples from treated mice 2 h (continued on next page…)

**Figure 4 (…continued)**
after 40 scrapes. 0.1 g of skin tissue were excised, rinsed in 1 μl PBS and homogenized on ice. The concentrations of TNF-$\alpha$ (A), IL-1$\beta$ (B), IL-6 (C), IL-10 (D), IL-12p70 (E), and IL-23 (F) in the supernatants were measured with ELISA and data were shown as mean + SD . Statistical comparisons were made between each group and the untreated group. ($n = 6$, \*, $p < 0.05$; \*\*, $p < 0.01$; \*\*\*, $p < 0.001$; One-way ANOVA with Dunnett's posttest.)

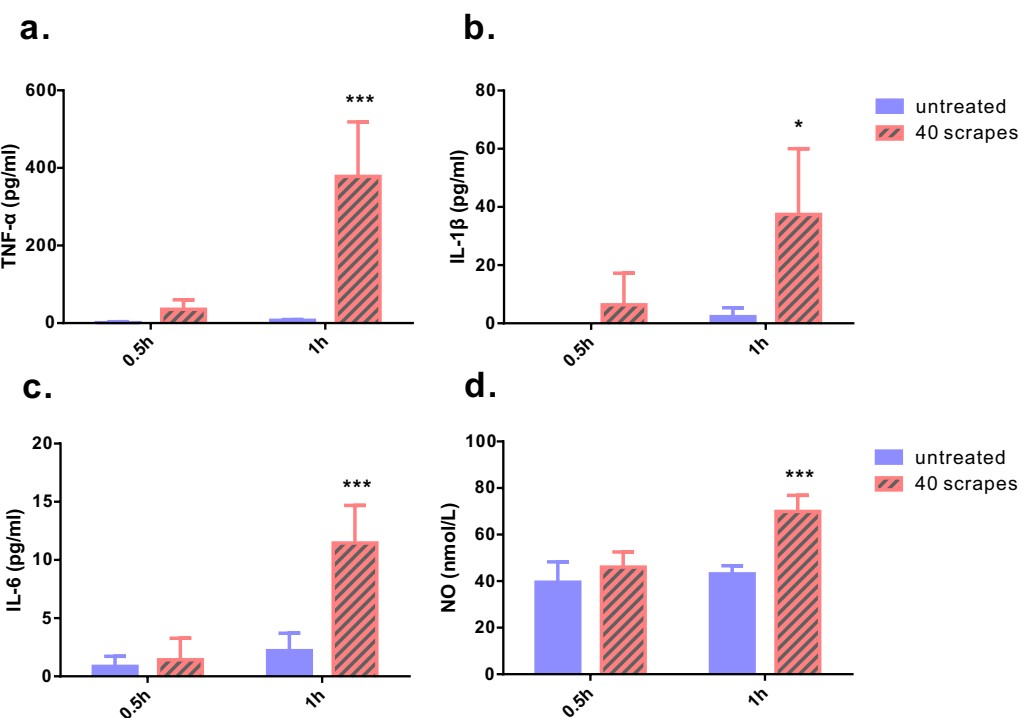

**Figure 5  Levels of pro-inflammatory cytokines and nitric oxide from serum samples.** Blood samples were taken 0.5 and 1 h after 40 scrapes Gua Sha treatment. Samples taken from untreated mice with identical blood-taking regime were as the control. The concentrations of TNF-$\alpha$ (A), IL-1$\beta$ (B), and IL-6 (C) were measured with ELISA and the content of nitric oxide was determined using the nitrate reductase method (D). Data were shown as mean + SD. Statistical comparisons were made between the treated and untreated groups. ($n = 6$, \*, $p < 0.05$; \*\*\*, $p < 0.001$; Two-way ANOVA with Bonferroni's posttest.)

three fold higher IgG titers, respectively after the second booster dose. In the untreated group, IgG2a induction was not very pronounced because half of the animals had titers below the detection limit. However remarkably, mice who received Gua Sha treatment before vaccination had elevated IgG2a titers, without any non-responders (Fig. 6E). The IgG1/IgG2a ratios of the treated groups after the second booster dose were lower than those of the untreated group (Fig. 6F), indicating that Gua Sha treatment of the skin prior to intradermal vaccination may lead to a more Th1-biased immune response.

## DISCUSSION

Gua Sha is one of the many old empirical practices, which are widely accepted as effective therapies, but the scientific understanding of its mechanisms is lacking. In traditional Chinese medical theory, it is briefly described as a therapy to help expel the toxicant,

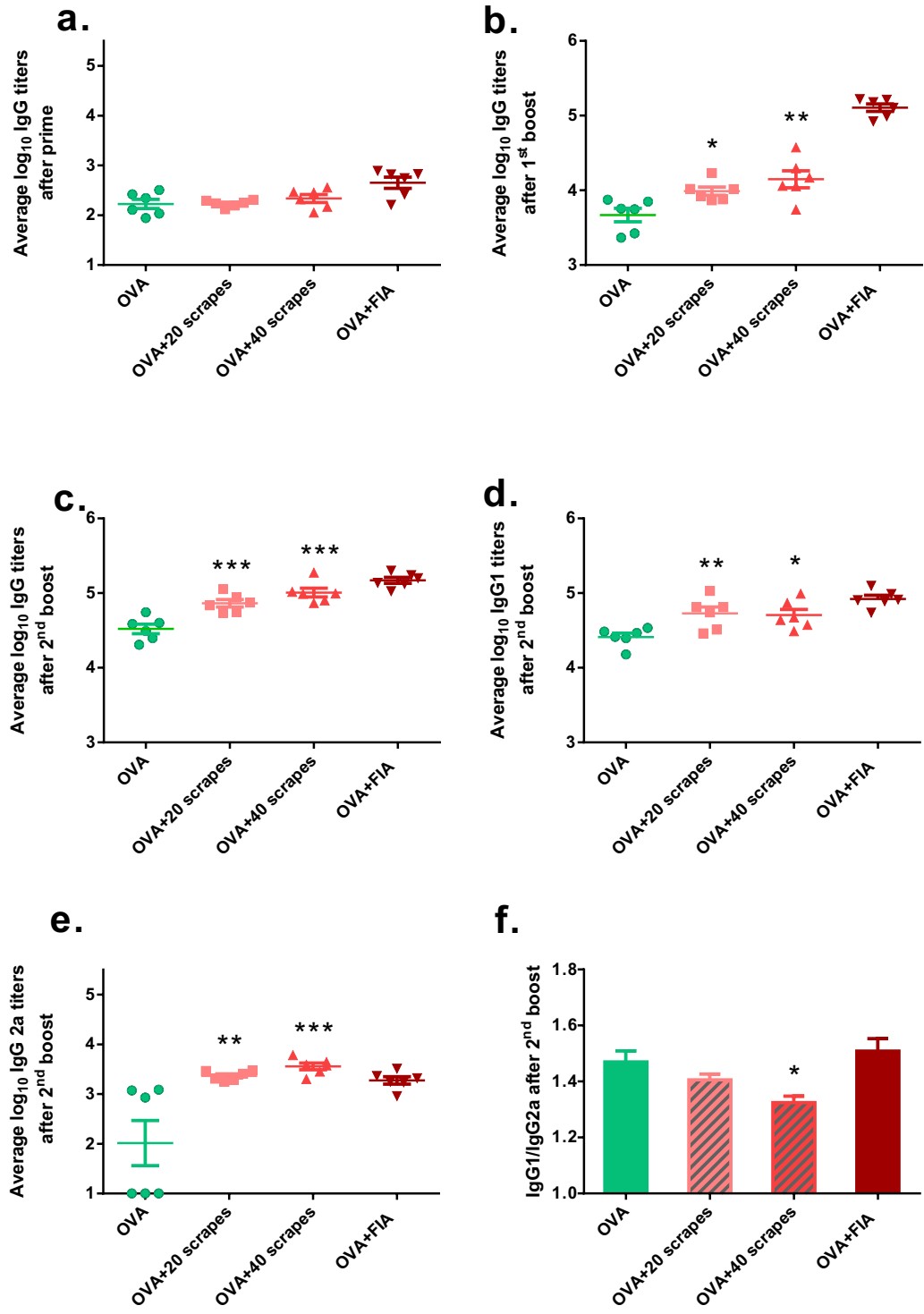

**Figure 6 Serum OVA-specific IgG subtype antibody titers after *i.d.* immunization.** Vaccination was performed 10 min after 20 or 40 scrapes at day 1, 21 and 42. Groups vaccinated with OVA and OVA + FIA in untreated mice were as the negative and positive controls, respectively. Sera were collected after prime, the first boost and the second boost (day 20, 41 and 55). (continued on next page...)

**Figure 6 (…continued)**
IgG (A, B & C), IgG1 (D) and IgG2a (E) were determined with ELISA. Data were shown as mean ± SD. OVA-specific IgG1/IgG2a ratios of individual mouse were calculated using IgG2a responders. Data were shown as mean + SD (F) ($n = 6$; *, $p < 0.05$; **, $p < 0.01$; ***, $p < 0.001$; One-way ANOVA with Dunnett's posttest.)

i.e., Sha, which cannot be cleared and excreted by physiological processes. However, Sha does not directly correspond to known biological substances. Articles about Gua Sha treatment published in the Western medical literature are mainly case reports describing its therapeutic impacts as well as complications, while only a handful of studies discuss its physiological impacts and underlying mechanisms (*Kwong et al., 2009*; *Lauche et al., 2012*; *Nielsen, 2009*). This effort is continued in the current study, and it is, to the best of our knowledge, for the first time, that Gua Sha is shown to be able to enhance the humoral immune response, generating higher antibody titers against vaccines applied *i.d.* on treated skin. It provides direct evidence that scratching on the skin as well as Gua Sha treatment can serve as a mechanical signal to enhance the immune surveillance function of the skin.

Although Gua Sha treatment is relatively simple, it generates multiple stimuli to the skin. The effect of Gua Sha on immunity may be a synergistic result attributed to the following factors. First, the mechanical press-stroke movement enhances surface microcirculation in the skin tissue and induces the expansion of capillaries, including the blood vessels and lymphatic vasculature (*Nielsen et al., 2007*). The increased blood and lymphatic flow leads to more rapid substance and cell exchange among blood, interstitial fluid, and lymphatic fluid. As the afferent lymphatic vessels are the route through which DCs migrate to the lymph nodes and lymphoid organs after antigen uptake, this, in turn, promotes the infiltration and migration of DCs and other immune active cells and accelerates the initialization of humoral immune response against the incoming antigens (*Wang & Oliver, 2010*). Second, the extravasation of blood and blood components, i.e., red blood cells and platelets, into the subcutis can induce multiple effects on immunity and inflammation (*McFadyen & Kaplan, 2015*), leading to a fluctuation in cytokine levels and initialization of the wound-healing process. Given the varied levels of cytokines at the local site after Gua Sha treatment, e.g., higher levels of Th1-biased immunopotentiators, namely, TNF-$\alpha$, IL-6, and IL-12 (*Afonso et al., 1994*; *Singh et al., 2011*; *Tovey & Lallemand, 2010*) and lower levels of the immunosuppressive IL-10, the antigen-presenting cells would be more rapidly activated and could lead to a stronger and more Th1-biased humoral response, as confirmed in the present study. In addition, Gua Sha generates temporary warming effects on the skin tissue, like short-term and local fever. We suggest that this thermal element of Gua Sha might also contribute in stimulating the innate and adaptive immune responses (*Evans, Repasky & Fisher, 2015*).

Gua Sha treatment creates controllable tissue damage and low-scale inflammation in the skin tissue, with a remarkable profile of changes in the cytokine levels. It prepares the surrounding skin tissues for possible pathogenic challenges and induces an immune defense exercise. Similar processes, which cause moderate tissue damage, such as the

use of a bifurcated needle for smallpox vaccination (*Rubin, 1980*), microneedle arrays (*Fernando et al., 2010*), tape-stripping (*Inoue & Aramaki, 2007*), non-ablative fractional laser (*Wang, Li & Wu, 2015*), low-frequency ultrasound (*Tezel et al., 2005*), and thermal ablation (*Garg et al., 2007*), have been reported to improve (trans)dermal vaccination to various extends. These are caused not only by tissue damage, but also skin barrier disruption, hence, increased antigen delivery. In contrast, in our study, *i.d.* vaccination of the same antigen dose clearly showed that tissue damage caused by Gua Sha treatment is the direct cause of the adjuvanticity observed. Although not determined in this study, Gua Sha treatment may as well compromise the skin barrier function. Standardized Gua Sha treatment with controllable impairment of the skin barrier may facilitate needle-free dermal vaccine delivery and improve its efficacy.

Gua Sha has been applied in human for centuries with strong safety records and it involves very simple instruments. However, Gua Sha therapies differing in the application force, directions, patterns, and inclusion of oils have been developed to treat different diseases and symptoms. A more elaborate and robust experimental design involving higher animals or human beings is, therefore, needed to unravel more about the underlying mechanisms. In practice, it could be applied prior to vaccination as a physical adjuvant, especially for those with poorly functioning immune systems. It needs to be pointed out that the ratios of treated/total skin area in the mice in this study are certainly higher than those in humans and, therefore, it may not be as effective as in mice. However, it is a safe, inexpensive, and versatile method, which can be employed separately or in combination with other biochemical adjuvants, without concerns about antigen–adjuvant or adjuvant–adjuvant interactions. As far as the mechanism of adjuvanticity is concerned, Gua Sha treatment may also boost the immune response to the vaccines delivered in the untreated skin area, subcutaneously or at the mucosal surface, and help convert the non-responders to certain vaccines, a domain that needs to be further studied.

## CONCLUSIONS

We conclude that Gua Sha treatment can improve the immunological functions of the skin and body by acting as a physical adjuvant, as shown by the increased ratio of immune active cells, variation in cytokine levels, and enhanced Th1-biased humoral immunity to intradermal vaccination at the site of treatment.

### Funding

This study was funded by the National Science Fund for Distinguished Young Scholars (81025019) and the National Basic Research Program of China (2012CB517603) for Prof. Junfeng Zhang; the Nanjing University State Key Laboratory of Pharmaceutical Biotechnology Independent Research Grant (ZZYJ-SN-201405) and the Project of Natural Science Foundation of Jiangsu Province (BK20161478) for Dr. Zhi Ding. The funders had no role in study design, data collection and analysis, decision to publish, or preparation of the manuscript.

## Grant Disclosures

The following grant information was disclosed by the authors:
National Science Fund for Distinguished Young Scholars: 81025019.
National Basic Research Program of China: 2012CB517603.
Nanjing University State Key Laboratory of Pharmaceutical Biotechnology Independent
Research Grant: ZZYJ-SN-201405.
Project of Natural Science Foundation of Jiangsu Province: BK20161478.

## Competing Interests

The authors declare there are no competing interests.

## Author Contributions

- Tingting Chen performed the experiments, analyzed the data, prepared figures and/or tables.
- Ninghua Liu performed the experiments, prepared figures and/or tables.
- Jinxuan Liu and Xiaoying Zhang performed the experiments.
- Zhen Huang performed the experiments, analyzed the data.
- Yuhui Zang contributed reagents/materials/analysis tools, wrote the paper.
- Jiangning Chen reviewed drafts of the paper.
- Lei Dong contributed reagents/materials/analysis tools, reviewed drafts of the paper.
- Junfeng Zhang conceived and designed the experiments, reviewed drafts of the paper.
- Zhi Ding conceived and designed the experiments, performed the experiments, analyzed the data, wrote the paper, prepared figures and/or tables.

## Animal Ethics

The following information was supplied relating to ethical approvals (i.e., approving body and any reference numbers):

All animals were handled in accordance with the Helsinki Declaration of 1975 (as revised in 2008) concerning animal rights, and the protocols were approved by the Institutional Animal Care and Use Committee of Model Animal Research Center of Nanjing University (Ref No. 20130015).

## Data Availability

The raw data has been supplied as Supplemental Files.

## Supplemental Information

Supplemental information for this article can be found online at http://dx.doi.org/10.7717/peerj.2451#supplemental-information.

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
