# Peer review of "Gua Sha, a press-stroke treatment of the skin, boosts the immune response to intradermal vaccination"

_PeerJ, doi:10.7717/peerj.2451_

## Round 0.1 · original submission · Major Revisions

· Academic Editor

Major Revisions

No additional Editor requirements. However, note that when Reviewer 2 mentions the following "Furthermore, to confidently conclude that DC and or macrophages are recruited to the area, these cells need to be enumerated (either by histology or better still, by purification)" I interpret that "by purification" means they want you to do flow cytometry on cells isolated from the skin to quantitate then number of DC in treated and untreated groups.

Please address all the reviewer comments.

Reviewer 1 ·

Basic reporting

The experiments were well conducted and clearly explained. Only a few points require clarification:
Fig1. At what time after Gua Sha treatment where the mice killed?
Fig2. It would help the reader if the different layers of the skin that can be seen in the pictures (epidermis, dermis and hypodermis/subcutaneous tissue) were labelled in one of the pictures as a reference. Also, arrowheads could be used to point to extravasated RBC in the dermis.
Fig 4. In fig legend, please specify that statistical comparisons are made between the “blank” group and the other groups. Also, I think it would be best to call the “blank” group “Naïve” or “untreated”, as “blank” may be confused with blank samples in the ELISA.
Check greek symbols in y-axis legend for TNF-a and IL-1b.

Text for fig 6. Fig 6a is not referred to in the text. Also, in lines 227-228 please indicate that you are referring to mice that did not receive Gua Sha or FIA.

Experimental design

The experiments are generally well designed, except for the claimed accumulation of DC and macrophages in the skin in Fig 3:
In that figure, DC are defined exclusively on the basis of CD11c expression. However, activated T cells can also express CD11c and T cells might accumulate in the skin after Gua Sha treatment. Therefore, can the authors co-stain for MHC-II to make sure they are looking at DC (CD11c+ MHC-II+)?
Also, to sustain the claim that DC or macrophages accumulate in the skin of treated mice, the immunohistological images in Fig 3 would need to be complemented with the quantification of these cells in the images.

Validity of the findings

The findings seem solid except for the claimed accumulation of DC in the skin of Gua Sha-treated mice.

Comments for the author

The manuscript is very interesting and presents convincing data on the pro-inflammatory effect of Gua Sha treatment. Only some extra information needs to be provided.

Reviewer 2 ·

Basic reporting

The authors investigated the effects of Gua Sha (skin scraping) on various immunological parameters. A number of interesting observations were made that warrant further examination. Essentially, the Gua Sha treatment appears to have immuno-stimulatory effects that could be harnessed in different settings.

The manuscript is logically set out but requires extensive English editing.

Experimental design

There are a number of significant concerns that the authors should address.
1) All figures must stipulate how many times an experiment was performed - this information is missing throughout manuscript. Reproducibility is a critical component of the analysis and interpretation of data.
2) Fig 3. Is supposed to show DC and macrophage movement in Gua Sha treated skin. The assumption is that DC/macrophages came into damaged tissue. However, the staining shown is difficult to interpret: inflammation/damage can affect non-specific background staining. I don’t necessary see an influx of DC – rather the matrix seems to bind the anti-CD11c mAb (and F4/80). The authors must include isotype control Ab staining. Furthermore, to confidently conclude that DC and or macrophages are recruited to the area, these cells need to be enumerated (either by histology or better still, by purification). Whilst enumerating the influx of cells maybe outside the scope of this paper, parallel (simultaneous) isotype control stained samples must be included – without this control the data cannot be interpreted.
3) Figure 4 – the column called blank – is this the naïve group of mice? If so, please change label. The authors shows statistically significance using ANOVA, but it is not clear if there is a statistically significant difference between naïve mice and scraped mice – perhaps a direct comparison between certain groups is needed.
Figure 5: The authors state that serum samples were taken prior to treatment and 0.5 an 1 h post treatment – the authors need to specify the blood volume collected – excessive blood loss could in itself have biological consequences. Also note, the group called blank – again – are these the naïve mice? – if so, please change label.
Most importantly, Figure 5 does not specify what group of mice were analysed – untreated, 20-scrapes, or 40-scrapes? I assume it was 40-scrapes. In order to account for the blood loss effect, the authors need to include naïve controls that undergo identical bleeding procedures (but are not scraped). These experiments must be done in parallel (not retrospectively).
Figure 6: It seems astonishing that there is a statistically significant difference between naïve mice vaccinate with OVA and scraped mice vaccine with OVA (Fig 6, b, c, d) – I question the validity of this statistical analysis.

Validity of the findings

The interpretation of the experiments are difficult in the absence of appropriate controls (all outlined above).

Comments for the author

Whilst it is interesting that Gua Sha appears to affect vaccine efficacy, the authors should highlight that they scrape a large surface area of the mouse which would exceed the surface area scraped on humans. Thus, the effects on mice are likely to be amplified. Nevertheless, an interesting observation.

---

## Round 0.2 · Minor Revisions

· Academic Editor

Minor Revisions

It is unnecessary to address reviewer 1's requirement to remove Figure 3c,d, but you might consider making the red line of untreated mice clearer so that it is obvious whether there is a difference between this and time points after treatment.

You must address the requirement from reviewer 2 to provide clear detail on how many times experiments were performed in each Figure.

Please also have the manuscript further edited by a native English speaker

Reviewer 1 ·

Basic reporting

The authors have addressed all my concerns

Experimental design

The experimental design is correct

Validity of the findings

In figure 3, I fail to see a difference in the proportion of F4/80+ cells in the histogram shown. The red and the blue lines (untreated vs 15 min) look identical for most of the graph, especially for the part of the histogram where the authors made their gate, and the differences between the red line and the green or orange lines are minimal. Is this a representative histogram? It is very surprising that the authors find a significant difference between the groups, especially since only 3 mice were used.

Comments for the author

I am not convinced that macrophage numbers changed in treated mice. I would consider removing this figure.

Reviewer 2 ·

Basic reporting

The authors have addressed all my comments, except one: They did not mention how many times each experiment was performed (in the Figure legends). Even if some groups have only been done once, the authors need to make this clear in the Figure legends.
Please note my original request was:
"All figures must stipulate how many times an experiment was performed - this information is missing throughout manuscript. Reproducibility is a critical component of the analysis and interpretation of data."

Experimental design

Good

Validity of the findings

Good

Comments for the author

Interesting findings - interesting implications.

---

## Round 0.3 · accepted · Accept

· Academic Editor

Accept

Thank you for your patience in revising this paper.

Reviewer 1 ·

Basic reporting

OK

Experimental design

OK

Validity of the findings

Figure 3 is clearer now, I can see the differences between the untreated and the treated groups